# Mapping of Ecological Vulnerability of Sea-Coastal Zones to Oil Spills: A Preliminary Method Applied to Kola Bay, the Barents Sea

**Anatoly Shavykin * and Andrey Karnatov**

Engineering Ecology Laboratory, Murmansk Marine Biological Institute of the Kola Science Center of the Russian Academy of Sciences (MMBI KSC RAS), 183010 Murmansk, Russia
* Correspondence: shavykin@mmbi.info

**Abstract:** Preparedness for oil spill response is a challenge for many coastal countries. Responders are unable to take effective action unless maps that indicate areas with different vulnerability to oil pollution are available. Such maps, developed in many countries, are usually based on calculations with rank (ordinal) values. However, arithmetic operations with them cannot be allowed. The article describes a method of constructing maps using metric values. The calculations take into account the biomass and the quantity of important biota components, especially significant socio-economic objects and protected areas. The biota distribution densities are represented in the identical units. The vulnerability factors are assessed based on the potential impact of spilled oil on biota, as well as its sensitivity and recoverability after disturbance. The proposed method takes into account the different sensitivity of biota inhabiting in the water column and on the sea surface. Oil vulnerability maps for Kola Bay using the proposed algorithm are presented.

**Keywords:** vulnerability maps of sea-coastal zones to oil; integrated vulnerability; sensitivity of biota; potential impact of oil; recoverability of biota; especially significant socio-economic objects; Kola Bay

## 1. Introduction

Sensitivity maps of coastline by environmental sensitivity index (ESI), susceptibility maps of shoreline and offshore, and vulnerability maps of sea-coastal zones to oil [1–3] are crucial tools in oil spill response (OSR). Sensitivity mapping procedures, which have a number of modifications, are well developed, and maps constructed by such procedures are widely distributed in many countries [3–12]. Also important is the approach to assessment of shoreline and offshore susceptibility to oil spills based on modeling of oil spill spread, taking into account bathymetric, meteorological, oceanographic, and geomorphological data [13–15]. Area hazard maps of oil spill spreading and diffusion are obtained as a result. Information about water area vulnerability (water column and bottom vulnerability) as for adjoined to coastline so for far from coastline—areas of different vulnerability—is required for OSR operations. Such water area vulnerability is defined by the presence of biota and social-economic objects. The vulnerability is different for various seasons. At the same time, it is not important where an oil spill happened—the vulnerability of each area supposes oil can reach it. In OSR operations, responders' actions are determined by (1) oil spill modeling results (shoreline and offshore susceptibility maps); (2) water area vulnerability maps; and (3) sensitivity maps of coastline by ESI. Thus, all these maps supplement each other and consequently could be used for minimization of ecological, economic, and other consequences of oil spills. Such maps could be useful for integrated management of coastal zones and resources, where resources assessment is important not only for coastline and inland [16–19] but for areas adjoined to coastline water areas. Biological and socio-economic resources

should be considered for these water areas with respect to ecological consequences of any natural and anthropogenic impacts. Our article is dedicated (to some degree) to this issue. However, approaches to mapping the vulnerability of sea-coastal zones have several issues; the main problems and the analyses of different methods are described in [20]. The main drawback of vulnerability mapping procedures is the use of rank (ordinal) values in calculations. As shown in [21–24] and demonstrated with specific examples in [20,25], arithmetic operations with ordinal values before their arithmetization are unacceptable. This article is a sequel to an earlier article [20].

Note that the approach to constructing vulnerability maps described below can also be applied to mapping hydroacoustic impact and impact of suspended mineral particles [26]. For this, the corresponding vulnerability coefficients should obviously be used. In general, such vulnerability maps of sea-coastal zones to various anthropogenic factors are important in assessing the environmental impact of offshore projects.

This article proposes a preliminary method for mapping vulnerability of sea-coastal zones to oil spill pollution (from source data to final maps) using Kola Bay of the Barents Sea as an example.

Throughout the article, we use the term "vulnerability maps" to refer to vulnerability maps of sea-coastal zones to oil spill pollution. The article gives an algorithm for constructing such maps and presents vulnerability maps of Kola Bay.

Some of our assumptions and estimated parameters that determine the biota vulnerability coefficients may be rather rough or not entirely accurate and may lack rigorous substantiation (which generally does not affect the workability of the proposed algorithm), but they help identify many unsolved methodological problems that should be addressed in further research. The finalization of vulnerability maps of Kola Bay also requires (apart from refining the biota vulnerability coefficients) that all interested parties (municipal and regional authorities, business entities, environmental organizations, etc.) approve the coefficient of priority protection of abiotic components, the contribution of biotic and abiotic components to total vulnerability, etc.

The proposed method is based on the recommendations of international organizations [1,3], other approaches developed in various countries [27], the experience of Russian specialists [10,28], and the results obtained by researchers of the Murmansk Marine Biological Institute (MMBI) [20,26,29–36].

## 2. Baselines and Definitions

The proposed method for mapping the vulnerability of sea-coastal zones to oil spill pollution, which can be applied for mapping vulnerability to the other anthropogenic impacts, was built on the following baselines:

The method should be fairly simple without complex formulas and should, if necessary, allow for the accounting of trophic and topical relations between the ecosystem components in the mapped zone by introducing appropriate coefficients.

At all stages of the integrated vulnerability calculation, ordinal (rank) values shall not be used, since this results in incorrectness [20–25,37]. At that, scores are permissible if they reflect the ratios between the values estimated by score on the metric ratio scale. After all arithmetic operations are completed, the intervals of the resulting integrated vulnerability range are ranked; however, these rank values should not be used in any subsequent calculations.

All maps are developed in the following scales: (a) strategic small-scale maps (if necessary) of the entire sea or a larger marine zone; (b) tactical medium scale maps of the coastal of the selected zone or a part thereof; (c) object-related large-scale maps of the most important natural features in the coastal zones [3]. All baseline information is prepared based on the selected scale.

Three groups of ecosystem features are taken into account in mapping: (a) important biota components (IBC), i.e., groups, subgroups, or species of biota inhabiting in the mapped zone; (b) especially significant social-economic objects (ESO), i.e., valuable natural features and socially and economically important facilities; (c) protected areas (PA), i.e., parts of the mapped water area that are classified as specially protected natural reservations.

A simple model of oil behavior during oil spill is adopted. It is assumed that the oil behavior generally depends on the density and the viscosity of oil, the hydrometeorological conditions of the mapped zone, and the structure of the littoral and the sublittoral shelves.

Biota vulnerability coefficients are calculated based on the potential impact of anthropogenic factors on the biota, the sensitivity of the biota, and its recoverability after the anthropogenic disturbance [27].

IBC vulnerability depends on: (a) the abundance of biota components (for the purpose of this article, "abundance" means total population or biomass within the mapped zone or area) represented in standard units (spec/km$^2$, kg/m$^2$, etc.); (b) the vulnerability coefficients of these components. The priority of ESO and PA protection is determined by the importance of the relevant part of the water area for the ecosystem and humans. It is, in fact, the priority protection coefficient (an analogue of the biota vulnerability coefficients).

Integrated vulnerability of the water area (water surface, water column, sea bottom) is determined by the sum of the IBC vulnerability and the priority of ESO and PA protection [20,32,36].

All maps are built using GIS applications.

The following definitions of the basic concepts are adopted in the proposed method [38].

*Mapped region* is an area within which all the input parameters of IBCs, ESOs, and PAs are taken into account for a selected scale. For tactical and strategic maps, this is the water area and the coastal zone within the boundaries of these maps. For object-related maps, it is either (a) a part or all of the impact area of a specific object disturbed by a federal oil spill emergency (more than 5000 tons) or a regional oil spill emergency (more than 500 tons), or (b) the area within the map.

The concepts of sensitivity and vulnerability are fundamentally different.

*Sensitivity* (of biota) is the ability of an organism to respond to changes in environmental factors; one should distinguish species sensitivity, age sensitivity, etc. [39].

*Vulnerability* (of water area, water column, sea bottom, or their combination) is a characteristic that describes possible consequences of the anthropogenic impact on the environment and the biota, leading to (a) disruption of the regional ecosystem and/or the ecological biota groups until their death; (b) changes in the natural habitat conditions of the biota; (c) functional failure of social and economic facilities and environmentally significant areas of the region.

In the proposed method, vulnerability maps can indicate relative and absolute vulnerability. *Relative integrated vulnerability maps* show the vulnerability of different parts of the water area for a particular season when each of them has its own range of vulnerability. These maps are directly intended for OSR planning.

*Absolute integrated vulnerability maps* are constructed for environmental purposes and integrated marine resources management as auxiliary maps for OSR plans. These maps are based on the annual range of the integrated vulnerability values and show the vulnerability of a given water area throughout the year, thus highlighting the most and the least vulnerable seasons in terms of the impact of oil spills. They also make it possible to compare individual sites by their vulnerability in different seasons.

## 3. Algorithm

*Collect the initial data*. It is assumed that the necessary data (geodatabases, field research, literature, reports of environmental impact assessment, engineering and environmental surveys, operational environmental monitoring, photos and video footage obtained by remote and aerial surveys, and other materials) has been collected to a greater or a lesser extent. If necessary, additional field research has been conducted to collect the missing information on the mapped region(s).

1.  *Determine and list the significant ecosystem features*, i.e., IBC, ESOs, and PAs. The relevant recommendations are described in [3]; they require certain refinement and regional clarifications.
2.  *Demarcate the seasons* for the mapped region. The boundaries of the seasons are determined for the biota components according to the constancy of the density of their distribution; for abiotic objects, they are determined according to their seasonal position. The boundaries of the seasons

for which integrated vulnerability maps are constructed are specified, taking into account the boundaries of all considered biotic and abiotic components.

3. *Construct the seasonal maps* of the distribution of the IBC abundance and the location of ESOs and PAs.

    3.1    *Construct the seasonal maps of the distribution of the IBC abundance* $B^{sg_j}$ (where $s$ is the season index; $g$ is the group index; $j$ is the subgroup/species index) in appropriate units, such as $g/m^2$, $spec/m^3$, tons per trawl hour, $spec/km^2$, and others.

    3.2    *Normalize the IBC maps.* The density of distribution of each $g_j$-th biota component is normalized to the total average annual abundance of the corresponding group $P^{yg}$ ($y$ indicates that the considered period is a year):

$$B^{(y)sg_j} = B^{sg_j}/P^{yg}, \tag{1}$$

where $P^{yg} = \langle \sum_q B_q^{sg_j} \times S_q \rangle$; $B^{(y)sg_j}$ is the fraction of the abundance of the $g_j$-th biota component (for season $s$) in the average annual abundance of the biota group in the mapped region per area unit within the polygons of the region; $S_q$ is the area of polygon $q$ for the corresponding $g_j$-th biota component; and triangular brackets indicate averaging over the year, taking into account the duration of the seasons.

    3.3    *Construct the seasonal maps of the location for ESOs* $C^{se}$ ($e$ is the ESO index), which are assigned the value 1.

    3.4    *Construct the seasonal maps of the location for PAs* $D^{sf}$ ($f$ is the PA index), which are assigned the value 1.

4. *Calculate the biota vulnerability coefficients* $V_b^g$ (the calculation is performed using the data on the properties of biota components, and it can be partly judgement-based) and make expert assessment of the priority protection coefficient for ESOs $V_c^e$ and PAs $V_d^f$ (taking into account ecological, economic, etc., importance of such objects to the humans or the ecosystem). All values of $V_b^g$, $V_c^e$, $V_d^f$ are on the metric ratio scale. A model of spilled oil behavior in the mapped water area is adopted at this stage. The vulnerability coefficients for the considered components largely depend on this model.

5. Construct the seasonal maps of IBC vulnerability and priority protection of ESOs and PAs.

    5.1    *For IBC*, "sum up" the maps of their distribution multiplied by the corresponding vulnerability coefficients $Y_b^s = \sum_{g_j} B^{(y)sg_j} \times V_b^{g_j}$, and normalize the resulting maps for each season: —for relative vulnerability maps $Y_b^{[s]s} = Y_b^s/(maxY_b^s$ for season $s)$; —for absolute vulnerability maps $Y_b^{[y]s} = Y_b^s/(maxY_b^y$ for a year$)$, where $Y_b^y$ is the set of all $Y_b^s$ values for all seasons. In each of these cases, after multiplying the obtained values by 100, we move, respectively, to the range of relative vulnerability values $minY_b^{[s]s} \div maxY_b^{[s]s}$ (=100 conditional units) and absolute vulnerability values $minY_b^{[y]s} \div maxY_b^{[y]s}$ as part of the total range $minY_b^{[y]y} \div maxY_b^{[y]y}$ (=100 conditional units). Indices $s$ and $y$ in square brackets denote a given map normalization performed at this stage (for relative or absolute vulnerability maps, respectively).

    5.2    *For ESOs*, "sum up" the maps of their location multiplied by the corresponding priority protection coefficients $Y_c^s = \sum_e C^{es} \times V_c^e$, and normalize the resulting maps for each season: —for relative vulnerability maps $Y_b^{[s]s} = Y_c^s/(maxY_b^s$ for season $s)$; —for absolute vulnerability maps $Y_b^{[y]s} = Y_b^s/(maxY_b^y$ for a year$)$, To construct the priority protection maps, we also move to the range of priority protection values—from minimum value to 100 conditional units.

　　5.3　　*For PAs*, calculate the maps and the formulas the same way as for ESOs.

6.　　Calculate the seasonal integrated vulnerability maps of the mapped region:　　　—for relative vulnerability maps $Y_{\Sigma}^{[s]s} = K_b \times Y_b^{[s]s} + K_c \times Y_c^{[s]s} + K_d \times Y_d^{[s]s}$;　　　—for absolute vulnerability maps $Y_{\Sigma}^{[y]s} = K_b \times Y_b^{[y]s} + K_c \times Y_b^{[y]s} + K_d \times Y_d^{[y]s}$, where $K_b$, $K_c$, $K_d$ are the coefficients of relative importance of the ecosystem components (IBCs, ESOs, and PAs), taking into account the need to maintain the ecosystem's normal activity and sustainable development of the region's economy.

　　Divide the range of obtained vulnerability values $Y_{\Sigma}^{[s]s}$ and $Y_{\Sigma}^{[y]s}$ for each season into five or three subranges; for the obtained vulnerability values $Y_{\Sigma}^{[s]s}$ for each season, there is its own range of values $min \div maxY_{\Sigma}^{[s]s}$, while for $Y_{\Sigma}^{[y]s}$, there is an overall range of values $min \div maxY_{\Sigma}^{[y]s}$. In any case, this division (classification) algorithm should be strict (reproducible) and properly described. The subranges with maximum integrated vulnerability have rank 5, and those with minimum vulnerability have rank 1 (if divided into three subranges, they have ranks 3 and 1, respectively). The polygons with different values (ranks) are marked in different colors on the resulting maps: rank 1—green, 2—light green, 3—yellow, 4—orange, 5—red (if divided into three subranges, the ranks are marked in green, yellow, and red). Indicate the selected colors of polygons with the corresponding ranks and the conditional units of integrated vulnerability in the map legend. The resulting seasonal maps of relative integrated vulnerability can be used in the OSR planning. Sites having ranks 4 and 5 are areas of priority protection (in the 3-rank scale, those are sites with rank 3). Sites having ranks 2 and 1 (sites with rank 1 in a 3-rank scale) are sacrificial and may, if necessary, accommodate spilled oil for subsequent collection. If there is a threat to neighboring sites with different vulnerabilities, the sites (area) with a higher rank should be protected first and, if necessary, those with a lower rank may be sacrificed. The absolute integrated vulnerability maps are intended for environmental purposes or as auxiliary visual materials for OSR operations. With the help of such maps, it is possible to compare the vulnerability of one site in different seasons.

　　In the next section, each step of the algorithm is explained by constructing a vulnerability map for Kola Bay.

## 4. Vulnerability Mapping of Kola Bay

　　1. *Ecosystem components and source data.* The IBCs that were taken into account when constructing vulnerability maps for Kola Bay were macrophytobenthos, zoobenthos, and birds. Ichthyofauna was ignored, since there are no data on its distribution in the bay due to the ban on trawling because of intense shipping traffic. Marine mammals were ignored, since they rarely enter the bay. Phyto- and zooplankton were also not considered due to rather high recovery [3]. Additionally, the groups were divided into subgroups, because spilled oil has different impacts on them, and their sensitivity and recoverability after disturbance are also not the same. In other sea-coastal areas, the division into groups/subgroups may be done otherwise.

　　The macrophytobenthos of the littoral and the sublittoral shelves of the bay were considered without division into subgroups. It was assumed that its distribution density is constant throughout the year. A detailed description and distribution maps of macrophytobenthos are presented in [40–43].

　　The zoobenthos were divided into three subgroups: (1) macrozoobenthos—bottom-dwelling invertebrates 1.5–30 mm in size; (2) mobile megazoobenthos—bottom-dwelling invertebrates over 30 mm in size and able to move and migrate over long distances; (3) non-mobile megazoobenthos—bottom-dwelling invertebrates over 30 mm in size and sessile or able to creep slowly. The distribution density (biomass) of macrozoobenthos and non-mobile megazoobenthos was taken as constant throughout the year, while mobile megazoobenthos was assumed to have two different densities for two seasons. The detailed description of the benthos and the corresponding distribution maps are given in [44,45].

All species of birds that inhabit Kola Bay were divided into three subgroups: (1) aquatic birds—species that spend most of their time on the water; (2) periwater larines—species that spend most of their time in flight or on the littoral shelf; (3) periwater sandpipers—species that live mainly in the coastal zone and in the littoral shelf and enter only shallow water. The detailed description of each subgroup and the seasonal distribution maps are given in [46–49].

The most significant ESOs in Kola Bay, such as port facilities and adjacent water areas (150 m), mouths of spawning rivers, crab breeding areas and larval development areas, and nesting and brooding sites of eider ducks, were also identified and reflected in the integrated vulnerability maps. The maps that indicate the location of these significant natural features and man-made facilities are given in [50].

There are no specially protected natural reservations in Kola Bay; therefore, PAs were not taken into account in vulnerability mapping.

2. *Season boundaries.* The boundaries of the seasons for the integrated vulnerability mapping were determined, taking into account the boundaries of the seasons of all the considered biotic and abiotic components. The detailed information on the adopted seasonal changes in the density of biota distribution and the ESOs in Kola Bay is given in [43–50]. Table 1 shows the boundaries of the seasons specified for the integrated vulnerability mapping of Kola Bay. The five seasons were determined in such a manner so that the distribution density of any biota component would not change in each adopted period (season).

**Table 1.** Division of year into seasons for mapping the integrated vulnerability of Kola Bay.

| Considered Objects | Months | | | | | | | | | | | |
|---|---|---|---|---|---|---|---|---|---|---|---|---|
| | I | II | III | IV | V | VI | VII | VIII | IX | X | XI | XII |
| Groups/subgroups of biota | | | | | | | | | | | | |
| Macrophytobenthos | | | | | | | | | | | | |
| Macrozoobenthos | | | | | | | | | | | | |
| Megazoobenthos (non-mobile) | | | | | | | | | | | | |
| Megazoobenthos (mobile) | | | | | | | | | | | | |
| Aquatic birds | | | | | | | | | | | | |
| Periwater (larines) birds | | | | | | | | | | | | |
| Periwater (sandpipers) birds | | | | | | | | | | | | |
| Especially significant objects | | | | | | | | | | | | |
| Port facilities and adjacent water areas | | | | | | | | | | | | |
| Mouths of spawning rivers | | | | | | | | | | | | |
| Crab breeding and larval development areas | | | | | | | | | | | | |
| Nesting and brooding sites of eider ducks | | | | | | | | | | | | |
| Seasons for Integral Vulnerability Mapping | Win | Early spring | | Spring | | Summer | | | Autumn | | Winter | |

*Note:* the periods with different spatial distribution of biota density and the periods of especially significant social-economic objects priority protection are marked with different colors.

3. *Initial seasonal maps of biota distribution and ESO location.* On the basis of information on the seasonal biota distribution, the initial IBC distribution density maps were constructed in the corresponding units of measurement accepted (phytobenthos in kg/m$^2$, zoobenthos in g/m$^2$, birds in spec/km$^2$). All the source data were normalized to the average annual abundance of the corresponding biota groups. The maps showing the ESO location, which were ranked 1, were based on the available literature and satellite images. All the initial maps and the information on their construction are given in [43–50] and online on the Geo-Information Portal of the Murmansk Region (web project Sensitivity and Vulnerability of Kola Bay to Oil Spills) [51].

4. *Biota vulnerability coefficients and ESO priority protection coefficients.* Type of oil and oil spill scenario are determined. The depth of oil distribution during surface spills may be different depending on the type of oil (its density), the water dynamics, and the position of the density jump layer. It was assumed that oil and oil products could be divided conditionally into three types (light, medium, and heavy oil), and vulnerability maps were constructed primarily for the average oil density (850–950 kg/m$^3$). In Kola Bay, almost along its entire length, the density jump layer was formed at a depth of 5–10 m during the year [52,53]. It prevents the penetration of oil into the depths, and spilled oil remains and

transforms in the layer from 0 to 5 m. This took into account that, due to the proximity of the coast (a small width of the bay) and hence the short duration of the oil slick on the surface, oil spilled in Kola Bay affects biota on the surface of the water, in the water column to a depth of about 5 m, in the littoral shelf, and in the sublittoral shelf to a depth of 5 m. To simplify the construction of vulnerability maps at this stage, it was assumed that oil does not sink to the bottom and does not pollute bottom sediments.

It is suggested to calculate the biota vulnerability coefficients ($V_b^g$) based on the approach proposed in [27]:

$$V_b^g = (E^g \times R^g)/S^g \tag{2}$$

where $E^g$ is the potential impact of spilled oil on the biota components taken into account, i.e., the probability of contact of organisms with oil (percent); $R^g$ is the biota recoverability, i.e., the ability of organisms to recover their population after the impact of oil spill pollution (years); $S^g$ is the sensitivity of the selected biota components (in this case, it was estimated in lethal values of the pollutant, and in calculations, it was normalized to the maximum permissible value of this substance in water).

In [27], the above formula has a different form—$V = (E \times S)/R$—since the values included in it are estimated not on a metric scale but in scores (ranks). Earlier, we also estimated these parameters in scores from 1 to 10 [30]. This method uses only metric values of the parameters on the ratio scale to perform arithmetic operations.

The potential impact of spilled oil ($E^g$) on biota means the probability of contact of organisms with oil and the nature of pollution. The value of $E^g$ depends on the duration of presence of the biota on the water surface, in the water column, or in the littoral shelf when oil is in these environments. At this stage, a time of real oil exposure on biota is not considered, and it is supposed that it is approximately the same for all species. If necessary, the time exposure could be taken into account by entering a corresponding coefficient.

For macrophytobenthos, the probability of contact with dispersed and dissolved oil in shallow water (0–5 m) and with oil film in the littoral shelf is rather high. Given that the mucous membrane protects macrophytes from sticking oil [54] and the thalli can be broken off [54,55], it was expertly assessed that $E^g = 70\%$ for macrophytes as compared with the possible impact of spilled oil on other subgroups of benthos (Table 2).

For zoobenthos subgroups, the value of $E^g$ varies. Mobile megazoobenthos such as crabs are able to avoid contamination zones [56–58]. Therefore, its $E^g$ value could be assumed to be minimal and was taken as 10%. The macrozoobenthos subgroup includes slow-moving and sedentary species that can burrow into the sand or close their shells, avoiding contamination [59–61]. Non-mobile megazoobenthos have fewer sedentary species that can hide in the thickets of macrophytes or leave the contaminated area than macrozoobenthos. The probability of the impact of spilled oil on these two groups was taken as 40 and 60%, respectively (Table 2).

Taking into account existing estimates of the impact of spilled oil on birds [46,62,63], it was assumed that, for aquatic birds, $E^g = 90\%$, and for periwater birds, $E^g = 35\%$.

Biota recoverability ($R^g$). The time that a species needs to recover a reduced population after the oil spill depends on the reproduction strategy, the population structure [58], the environmental conditions, the duration of exposure to oil pollution, the degree of population damage, and other factors.

For macrophytes of the Murman shore, it was experimentally determined that fucoids and common plant species are able to recover their biomass in a 1 m$^2$ littoral strip in the sheltered area in 4 years [64]. On the open shores, it is significantly reduced [65], hence it was expertly assessed that the recovery time of macrophytes population is 5 years.

The recovery time of macro- and megazoobenthos populations in the Barents Sea (completely disturbed communities) after bottom trawling is at least 5 years [66]; when calculating the damage to fish stocks, 3 years [67]. We assumed the average recovery time of 4 years for macrozoobenthos and non-mobile megazoobenthos. After a one-time extermination of 10% of individuals in a crab population, recovery to the initial level occurs in 3.5 years; after extermination of 20%, recovery occurs

in 4.2 years [personal communication by bentologist A.G. Dvoretsky of Murmansk Marine Biological Institute]. Taking into account these data, the $R^g$ values given in Table 2 were accepted for zoobenthos.

**Table 2.** Vulnerability parameter estimates ($S^g$, $E^g$, $R^g$) and final vulnerability coefficients ($V_b^g$) for Kola Bay biota.

| Biota | $LC_{50}$, mg/L | $LT_{50}$, μm | $S^g$ | $E^g$, % | $R^g$, Year | $V_b^g \times 100$ |
|---|---|---|---|---|---|---|
| **MPC = 0.05 mg/L** | | | | | | |
| Macrophytobenthos | 550 (100–1000) | | 11,000 | 70 | 5 | 3.2 |
| Macrozoobenthos: | 290 | | 5800 | 40 | 4 | 2.8 |
| polychaete | (10–100) | | | | | |
| bivalves | (50–500) | | | | | |
| gastropods | (100–1000) | | | | | |
| Non-mobile Megazoobenthos: | 410 | | 8200 | 60 | 4 | 2.9 |
| bivalves | (50–500) | | | | | |
| gastropods | (100–1000) | | | | | |
| Mobile Megazoobenthos—Crustaceans | 55 (10–100) | | 1100 | 10 | 3.5 | 3.2 |
| **MPT = 0.04 μm** | | | | | | |
| Aquatic birds | | 25 | 625 | 90 | 2 | 28.8 |
| Periwater (larines) birds | | 25 | 625 | 35 | 3 | 16.8 |
| Periwater (sandpipers) birds | | 25 | 625 | 35 | 3 | 16.8 |

*Note:* Numbers without brackets indicate the average value; numbers in brackets indicate the range of values; $S^g = LC_{50}/MPC$ for benthos; $S^g = LT_{50}/MPT$ for birds; $LC_{50}$—lethal concentration of a substance exposure that causes lethal effects in 50% of a standard population of organisms in a given time; $LT_{50}$—lethal thickness of an oil film that causes lethal effects in 50% of a standard population of organisms in a given time of location on the water surface with this film (this term needs further improvement and definition); MPC—maximum permissible concentration; MPT—maximum permissible thickness.

The recovery time of different bird populations noticeably varies. It can take 3–10 years to restore a population of Uria species after a one-time extermination of 10% of adult individuals and nestlings; to compare, it takes 2–6 years after a one-time extermination of 5% [68]. After the Exxon Valdez oil spill in 1989, most populations, including Uria communities, were considered "recovered" in 1.5 and 2.5 years. The recovery periods for other bird species were much longer, 3–9 years [69]. Aquatic birds have a higher reproductive potential than periwater birds. The share of all birds of Kola Bay [48] is less than 0.1% of the total number of bird populations in the Barents Sea region [70]. According to our tentative estimates, in case of oil spill in Kola Bay, no more than 5–10% of the total number of birds inhabiting the bay can die per season (that is no more than 1700 individuals in the summer). With that in mind, it was assumed that the recovery time for aquatic bird populations of the bay is 2 years, and for larines and sandpipers, about 3 years (Table 2).

Biota sensitivity to oil ($S^g$). The toxicity of petroleum hydrocarbons for biota varies tremendously, since the toxicity level is determined by many factors [54]. Therefore, for groups/subgroups of aquatic organisms, we proceeded from the average values of $LC_{50}$ ranges (assuming at this stage that their distribution is symmetrical; see Table 2). Another possible approach is based on lethal loading rate $LL_{50}$ [71,72] [$LL_{50}$—loading rate of a test substance (in dilution water), which causes lethal effects in 50% of the exposed population of organisms in a given time]. In this work, we considered the impact of crude oil (having a density of 850–950 kg/m$^3$) with 10% content of aromatic fractions; the $LC_{50}$ ranges for this substance are presented in [73].

The marine avifauna is probably the most sensitive component of marine ecosystems. Short-term contact with oil dramatically reduces the insulation power of feather cover in birds, leads to overcooling, and often ends in death. When estimating sensitivity, it should be borne in mind that, while the inhabitants of the water column (ichthyoplankton, fish, macrophytes, benthos larvae, partly marine mammals, etc.) are affected by oil concentration in water, the lethal factor for birds is the oil film on the water surface or the littoral shelf.

Next, we proceeded from a simplified simulation of the impact of spilled oil pollution on the marine avifauna. It was assumed that the only lethal factor acting on birds is oil film or individual oil splits on the water surface. The ingestion of oil with food or during preening and the pollution of nests (eggs and nestlings) at this stage were not taken into account.

Article [69] presents data on the impact of spilled oil on birds from several sources. For example, 0.1 g/m$^2$ (0.25 μm) oil film is too thin to cause high mortality in birds resting or swimming in it, while 1 g/m$^2$ (0.8 μm) oil film is 100% lethal. Article [69] argues that this contradicts the information of other authors who do not consider spills with film thickness of less than 1 μm harmful for birds. Another study [74] showed that thin oil films (0.1 and 0.3 μm) can affect the structure of feathers, but it is not clear whether they can lead to significant consequences for exposed birds.

Report [75] states that oil film thickness of 0.04 μm is an evident minimum threshold that can be used to determine the impact on socio-economic resources (for example, the closure of fisheries). This minimum thickness applies to oil slicks with barely visible colorless or silver gloss [76]. In that paper, it is assumed that 10 μm is the threshold for biological impact on the water surface (on birds). The threshold values of oil film thickness, which have a very negative effect on birds, are presented in few publications. Thus, work [77] with reference to [78] gives the following data—the threshold film thickness, which has a clearly manifested negative effect on birds, is 25 μm (25 mL/m$^2$), and any film thickness exceeding 25 μm is harmful to birds that get exposed to it. Taking into account the non-rigorous justification presented in [79] that 10 μm films are lethal for birds, we accepted the data in [77] and proceeded from the assumption that 25 μm films have 50% lethal effect (LT$_{50}$) on birds, i.e., LT$_{50}$ is similar to LC$_{50}$ for organisms inhabiting the water column.

However, it is impossible to use LC$_{50}$ for aquatic organisms and LT$_{50}$ for birds for joint calculations, since they are expressed in different units. Therefore, it was proposed to use LC$_{50}$/MPC ratio (we took MPC = 0.05 mg/L [80]) for the organisms inhabiting the water column, and LT$_{50}$/MPT ration (we took MPT = 0.04 μm [75]) for organisms interacting with the water surface as the $S^g$ value (Table 2). Thus, $S^g$ was expressed in the same units, i.e., in fractions of the standard lethal characteristics of the pollutant (LC$_{50}$ or LT$_{50}$).

The prior protection coefficients for the considered ESOs (port facilities and adjacent water areas, mouths of spawning rivers, crab breeding areas and larval development areas, nesting and brooding sites of eider ducks) were preliminarily estimated and taken as $V_c^e = 1$, since, according to our assessment, their significance for the ecosystem and humans is about the same [38].

5. *Construction of IBC vulnerability maps and ESO priority protection maps*. To produce the seasonal IBC vulnerability map, we "summed up" the IBC distribution maps (macrophytobenthos, zoobenthos, and birds) multiplied by the corresponding vulnerability coefficients (see Item 5.1 Section 3). These maps were normalized to the maximum values of vulnerability for the corresponding season (winter, early spring, spring, summer, or autumn) to receive the seasonal maps of relative biota vulnerability. After normalizing to the maximum value of vulnerability for the year, we received the seasonal maps of absolute biota vulnerability.

The ESO priority protection maps were constructed in the same way. The ESO location maps for each of the seasons were "summed up" and multiplied by the respective priority protection coefficients (see Item 5.2 Section 3). To construct the relative priority protection maps, the obtained values were normalized to the maximum values of protection priority for the corresponding season. To construct the absolute priority protection maps, the obtained values were normalized to the maximum values of protection priority for the entire year.

Thus, we mapped the IBC vulnerability and the ESO priority protection for five seasons in Kola Bay.

6. *Calculation of seasonal integrated vulnerability maps*. To obtain the relative integrated vulnerability maps of Kola Bay, the relative IBC vulnerability maps and the relative ESO priority protection maps for each season were "summed up" using formula $Y_\Sigma^{[s]s} = K_b \times Y_b^{[s]s} + K_c \times Y_c^{[s]s}$. The range of obtained vulnerability values for each season ($min \div max Y_\Sigma^{[s]s}$) was divided into five subranges. The subranges

with maximum integrated vulnerability received rank 5; those with minimum integrated vulnerability received rank 1.

To obtain the absolute integrated vulnerability maps, the absolute IBC vulnerability maps and the absolute ESO priority protection maps for each season were "summed up" using formula $Y_{\Sigma}^{[y]s} = K_b \times Y_b^{[y]s} + K_c \times Y_b^{[y]s}$. The united range of obtained vulnerability values for all seasons $(min \div max Y_{\Sigma}^{[y]s})$ was also divided into five subranges.

Coefficients $K_b$ and $K_c$ were expertly assessed as 2 and 1, respectively, since it was assumed that the contribution of IBCs to the general vulnerability of Kola Bay is twice as significant as that of ESOs [38]. The coefficient $K_d$ was not assessed due PA is absent in the bay.

## 5. Discussion of Integrated Vulnerability Maps of Kola Bay

Figures 1–4 show the integrated vulnerability maps of Kola Bay water area constructed for the summer season by the algorithm described above. All the calculations were performed in ArcMap 10.0. Tactical maps (1:150,000) of relative (Figures 1 and 2) and absolute (Figure 1) vulnerability are presented. Figures 3 and 4 show the object-related maps (1:25,000) of relative vulnerability for Region 3, the boundaries of which are shown on tactical maps. Strategic maps were not developed for Kola Bay, since a tactical map is enough for its size in our approach.

The integrated vulnerability range $min Y_{\Sigma} \div max Y_{\Sigma}$ and its division into subranges are presented on all the maps in the upper left corner; the abscissa axis is the scale of $Y_{\Sigma}$ values, and the ordinate axis is the number of polygons with the corresponding $Y_{\Sigma}$ values. The methods for dividing the range of integrated vulnerability values into subranges (classification) may generally be different. In our case, it is the method of equal intervals (Figures 1 and 3) and the method of equal-area division (Figures 2 and 4). The boundaries of areas with different vulnerability ranks and, in general, the entire resulting vulnerability map for the mapped region depend on the chosen classifications of the integrated vulnerability range to subranges.

Tactical maps 1:150,000. To apply the method of equal intervals, the entire range of integrated vulnerability values is divided into equal subranges from $min Y_{\Sigma}$ to $max Y_{\Sigma}$. This approach is recommended in GIS guides as the most suitable one for known ranges of values. It focuses on the value of the attribute relative to other values. The vulnerable areas obtained in this way do not depend on their location, area, and quantity, and the maps are fully consistent with the requirements of International Maritime Organization and International Petroleum Industry Environmental Conservation Association (IMO, IPIECA) that vulnerability maps covering a large area should indicate areas particularly sensitive to accidental oil spills [1]. Tactical maps can be used as a general planning and response tool, where individual vulnerable areas can be shown and users can be guided by them during OSR in case of relatively large oil spills. In Kola Bay, the most vulnerable (with ranks 4 and 5; Figure 1) are areas in the northern part of the northern bend of the bay along the western and the eastern shores (rank 5) and small areas in Pitkova Bay and in the mouth of Srednyaya Bay (rank 4), which should be protected first. In case of an oil spill in the southern or the middle bend of the bay, OSR operations should focus on vulnerable areas with ranks 3 and 2.

Let us note one downside of the method of equal intervals. A map constructed this way may end up having only one small area with high vulnerability (rank 5), and the rest of the large mapped area will be ranked 1 as least vulnerable (refer to Figure 1 for a similar situation). In such a case, the responders will have neither targets nor limits of what must be saved (except for one small area, for example, on the edge of the map) and what can be sacrificial. Hence the responder's actions in such a mapped region will not minimize the damage from the spill and the OSR operations. Therefore, when constructing maps in such cases, it may be necessary to use some other classification principle.

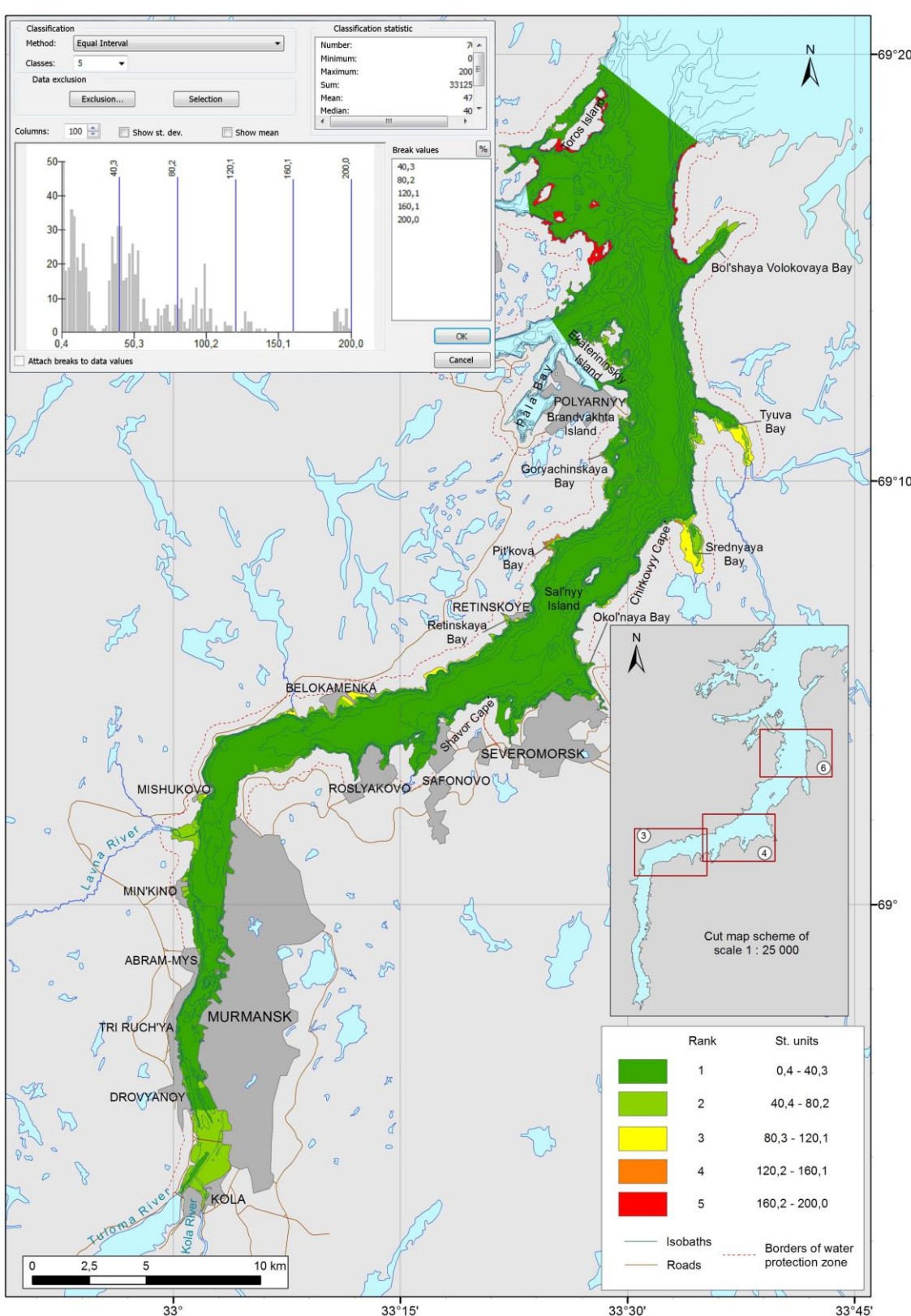

**Figure 1.** Map of relative and absolute integrated vulnerability of the Kola Bay water area to spilled oil in summer (VI–VIII). The classification is performed by the method of equal intervals [81] (Figure 13.4, p. 324 and Figure 13.9, p. 330).

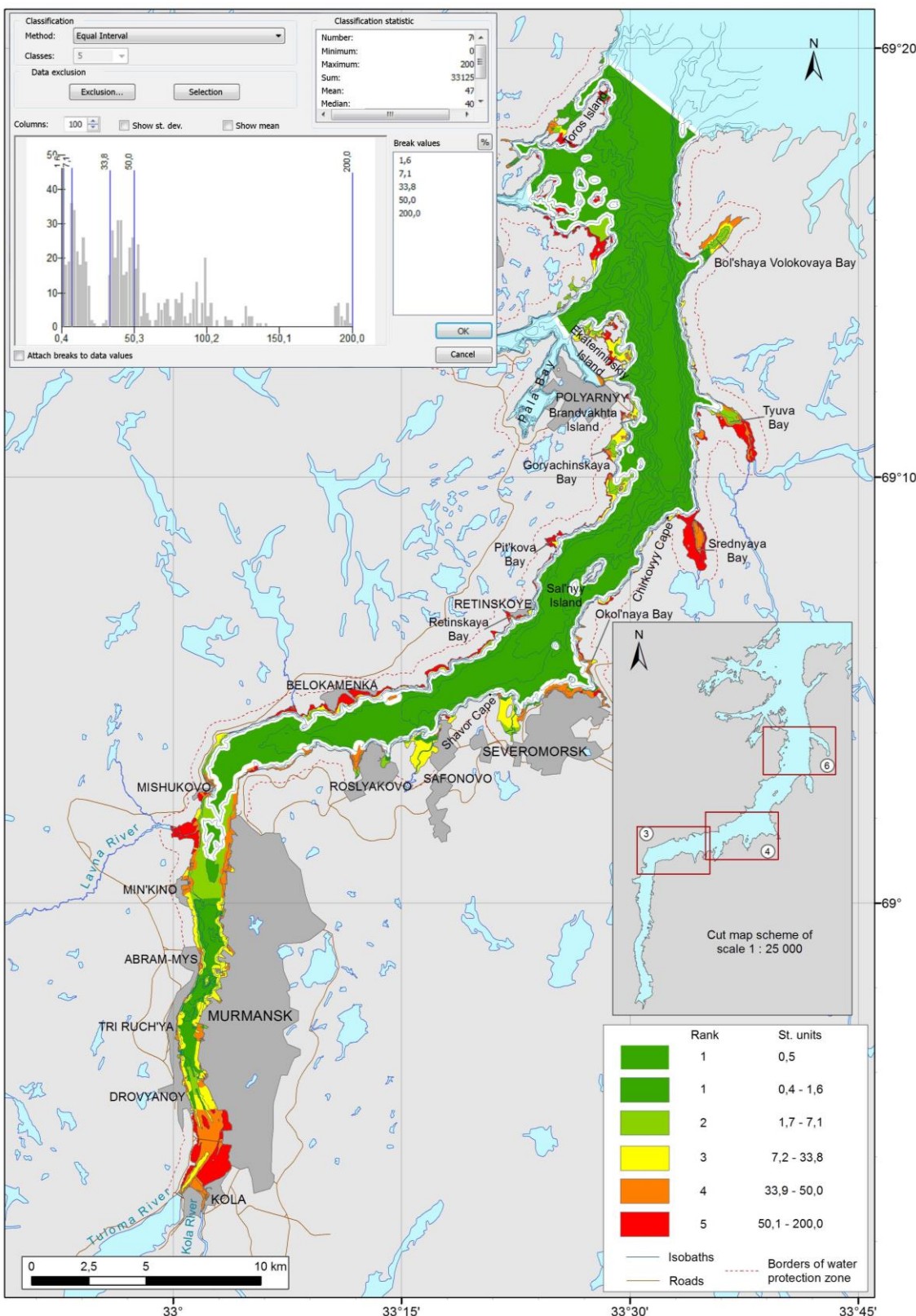

**Figure 2.** Map of relative integrated vulnerability of the Kola Bay water area to spilled oil in summer (VI–VIII). The classification is performed by the method of equal-area division [82] (Figure D.4, p. 477). The largest and least vulnerable area in the central part of Kola Bay is marked with white lines (boundaries of polygon).

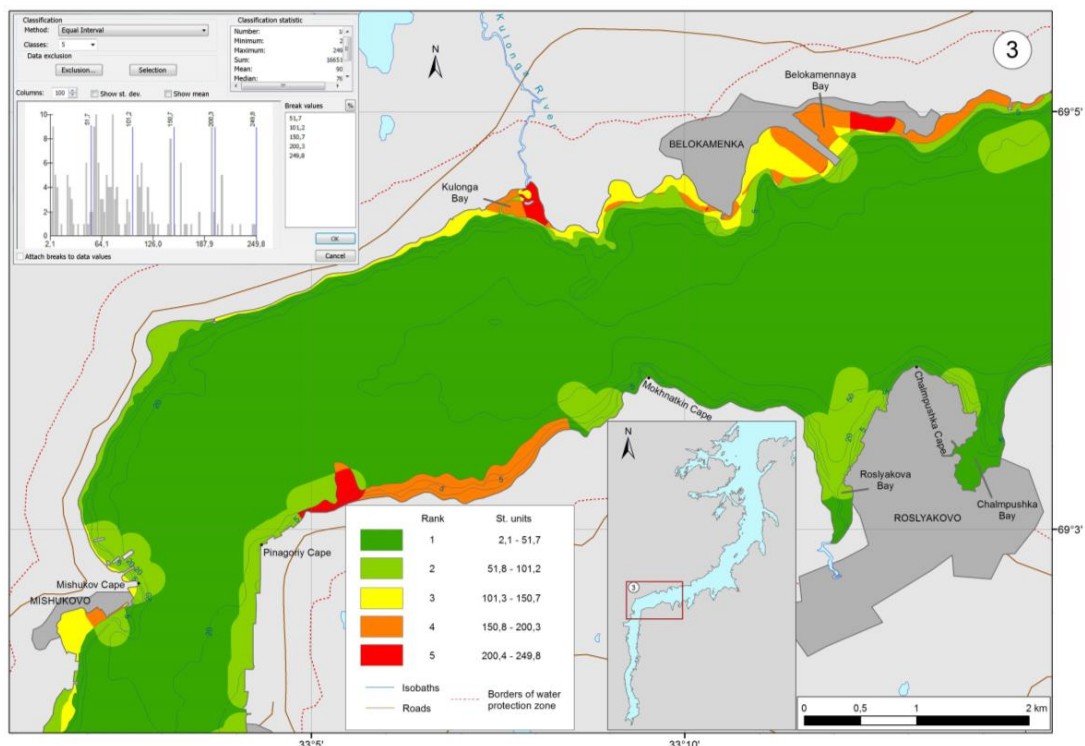

**Figure 3.** Map of relative integrated vulnerability of Region 3 in Kola Bay to spilled oil in summer (VI–VIII). The classification is performed by the method of equal intervals [81] (Figure 13.14, p. 336).

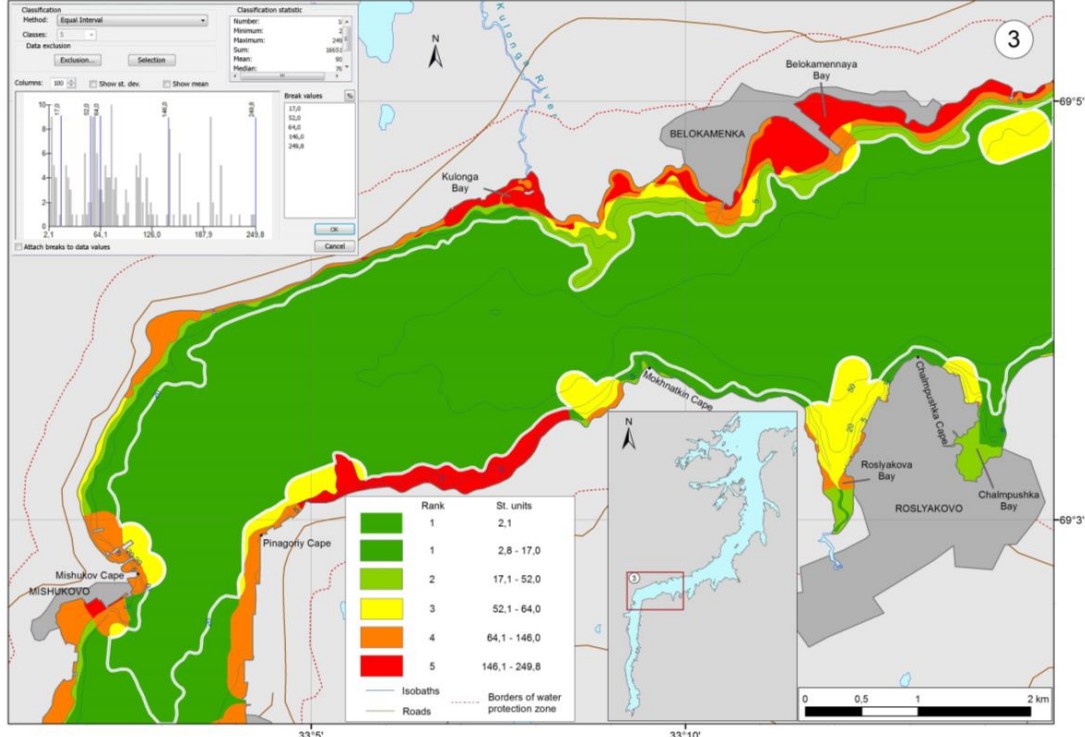

**Figure 4.** Map of the relative integrated of Region 3 in Kola Bay to spilled oil in summer (VI–VIII). The classification is performed by the method of equal-area division. The largest and least vulnerable area in the central part of Kola Bay is marked with white lines (boundaries of polygon) [82] (Figure D.9, p. 482).

There is another approach to the final classification of integrated vulnerability when the goal is to present maps where zones of high and low vulnerability alternate on each relatively small area (1–2 km). This will allow the responders to orientate themselves in each small area (in our case, in any part of the bay) to focus on zones of high or low vulnerability and thus to minimize damage from the oil spill and the OSR operations. Such approach to classification can be, for example, the method of "equal area" division. The range of values $Y_\Sigma$ ($minY_\Sigma \div maxY_\Sigma$) is divided into subranges, each of which corresponds to an approximately equal total area of all areas (polygons) within it that have a particular vulnerability.

With such division, the largest polygon in the area can be excluded from consideration if it significantly (approximately 10 or more times) exceeds the area of any other polygon. In this case, excluded is an area with minimal (or close to minimal) vulnerability, the boundaries of which are 100 m or more from the coast (its boundaries are marked with a white line in Figure 2). All other areas of the water area are divided into five groups with sub-ranges of vulnerability from very high (rank 5) to very low (rank 1). For each of these groups, the summed area of all the areas (polygons) within them has approximately the same value. On the map constructed using this method, areas of high vulnerability (ranks 4 and 5) are present in all areas of the bay and alternate with areas of less vulnerability. Other ways of classifying the values $Y_\Sigma$ ($minY_\Sigma \div maxY_\Sigma$) are possible, but all the maps of the mapped area should be constructed by a single algorithm. It should help the responders "see" the zones of relatively high (ranks 4 and 5) and relatively low (ranks 1 and 2) vulnerability in any small area.

However, it is worth noting a number of limitations of this approach. Thus, in areas of the same range, for example, with very low (or very high) vulnerability, there can be both very minimal and close to maximum vulnerability values. This will not be apparent to the responders who use the map, and it may lead to aggravated damage and, as a consequence, to inability to distinguish the most vulnerable areas from the least vulnerable ones in the whole area. For example, with the same values of vulnerability but different distribution of polygons (i.e., their areas), the same value of vulnerability can fall into rank 1 (least vulnerable) in one case and to rank 5 (most vulnerable) in another. Also, this approach is not usable for construction of absolute vulnerability maps. There are several other reasons for possible errors. This classification method is hardly applicable if the distribution of vulnerability follows one geographical direction.

According to IMO and IPIECA recommendations, object-related vulnerability maps should be used for small spills and at the completion stage of OSR operation to larger oil spills by mid-level team leaders and responders at the site of the spill. At that, object-related maps are prepared only for particularly important regions, such as the most sensitive areas, specially protected natural territories, oil terminals, bays with important port facilities, and others. In our case, the object-related maps were built for three of nine regions in Kola Bay (the total number of mapped regions was chosen due to the need to map the entire bay at of the scale of 1:25,000). Here, we present only the maps for Region 3 (Figures 3 and 4).

In general, in all the maps, the polygons of high and medium vulnerability appeared to be located near the shores of the bay due to the distribution of biota (benthic organisms are taken into account to a depth of 5 m; the distribution of birds is mainly confined to the coastline) and the chosen model of average density oil behavior. The distribution of vulnerable areas may differ when building vulnerability maps for other types of oil. For example, in case of heavy oil, it will be necessary to take into account the entire bottom flora and fauna.

To compare the vulnerability of individual areas in Kola Bay by season, the absolute integrated vulnerability maps were calculated using the method of equal intervals. The tactical absolute vulnerability map for the summer season (Figure 1) repeats the relative vulnerability map, since they have the same range of integrated vulnerability. In all other seasons, almost the entire water area of the bay has a minimum vulnerability, and only separate small areas have ranks 2 and 3.

## 6. Conclusions

In this article, we devised a method for constructing vulnerability maps of sea-coastal areas that can be exposed to spill oil pollution. The method is based entirely on the metric approach, which makes it impossible to use ranks in arithmetic operations when calculating maps and thus makes them more accurate and methodologically correct.

The method includes a solution to the problem of calculating vulnerability maps if the abundance and the biomass of various ecological biota components are presented in different units (g/m$^2$, kg/m$^2$, spec/km$^2$, etc.). The same approach can be applied to other components of biota (for example, ichthyoplankton or fish) if their initial distributions are given in spec/m$^3$, tons per trawl hour, etc.

The article proposes a solution to the joint consideration of the vulnerability of biota that dwells in contact with the water surface (birds) and the biota inhabiting the water column (macrophytobenthos and zoobenthos). This approach is based on different normalizing of the sensitivity of biota components that inhabit the water column (LC$_{50}$ or LL$_{50}$ values per the maximum permissible concentration of the pollutant) and regularly contact the water surface (LT$_{50}$ values per the maximum permissible thickness of oil film). The coefficients of vulnerability to average density oil for the ecological components of the Kola Bay biota were calculated (taking into account potential impact of oil, sensitivity to oil, and recoverability after disturbance). At this stage, these calculations are preliminary and require further study to justify the vulnerability coefficients. In addition, the baseline data for calculation of vulnerability coefficients may vary in different regions.

The integrated vulnerability maps of Kola Bay were constructed using different classification methods for the total values of integrated vulnerability. The presented relative vulnerability maps indicate the most vulnerable areas that require the most attention when planning and conducting OSR operations. The presented absolute vulnerability maps can be used as auxiliary materials for OSR operations, environmental monitoring, etc.

The results presented in the article are preliminary. We hope to receive feedback on the described method and possibly on approaches to solving the problems raised here.

**Author Contributions:** A.S. and A.K. developed the presented method and wrote the paper. All authors read and approved the final manuscript.

**Funding:** This research was funded by Ministry of Science and Higher Education of the Russian Federation (State Registration No. AAAA-A18-118030690062-0).

**Acknowledgments:** This paper is based on the results of scientific research performed for the state assignment of MMBI KSC RAS "Impact of climate factors, chemical and radiation pollution on Arctic marine ecosystems in integrated nature management condition" (State Registration No. AAAA-A18-118030690062-0).

**Conflicts of Interest:** The authors declare no conflict of interest.

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
