# Peer review of "Mapping of Ecological Vulnerability of Sea-Coastal Zones to Oil Spills: A Preliminary Method Applied to Kola Bay, the Barents Sea"

_jmse, doi:10.3390/jmse7070216_

Reviewer 1 Report

The paper is pleasant to follow and is very well written. I found very nice the five figures and the references match perfectly the text.

The Scientific interest is high because the proposed method has the objective of simplicity. That renders the maps useful for another public than the solely scientific community (OSR…). To prove the simplicity of the method (preliminary method) may be some improvements may be search.

Vulnerability mapping is a tremendous topic, and I encourage this paper to find its place in the worldwide works done on ESI.

Generally, the areas of research are those the authors’ institutes. Consequently, the methods proposed full fill well the specificities. I mention, Fattal for its works on the Atlantic and Kokkonen on the Baltic (Gulf of Finland)

·         FATTAL P., MAANAN M., TILLIER I., et al., “Coastal vulnerability to oil spill pollution: the case of Noirmoutier island (France)”, J. of Coastal Res., vol. 26, no. 5, p. 879-887, 2010.

·         KOKKONEN T., IHAKSI T., JOLMA A., et al., “Dynamic mapping of nature values to support prioritization of coastal oil combating”, Environmental Modelling & Software, vol. 25, no. 2, p. 248-257, 2010.

Two general comments:

- from the financial perspective, the ESI topic is related to IOPC fund . ESI map can be superposed with those of the financial compensations and a correlation should be obtained.

- from the juridical point of view:  the environmental damage enters into force in the court decisions. ESI maps might be used by court. The objective of simple reading of vulnerability maps could serve to the judge as well to the responders (line 55).

 After these generalities, I suggest the following:

The constants Kb Kc Kd are chosen as the same for seasons and year. Is this hypothesis linked to obtain a simple method ?

Time exposure of the different species to oil is not considered. It appears as an hypothesis which could be mentioned. It is closed to ‘duration of presence’ line 283.

Table 2 explains the equation 2 and illustrates the simplicity objective of the method (relation input-output). I suggest a similar approach with a new table explaining equation 1, or the expressions of items 5-6 paragraph 3. This table could mention some notations of the method.

At item 6 of paragraph 4, I suggest to recall the absence of PA in the bay (Kd=0) or to precise it at lines 237-238.

Line 73 proposes 3 kinds of map and only two kinds are presented (lines 408-409). The avoidance of a strategic map should be explained. A reference of ArcMap 10.0 may be added but it is not mandatory.

I understand that LC LL and LT are Lethal Concentration, Lethal Loading Lethal Thickness (of oil) and that the subscript 50 means lethal for 50% of a species. Can these definitions be given explicitly together (lines 332-335 & 525-527)? I suggest to place that before the first mention at line 291 before table 2.

Reviewer 2 Report

Dear authors

I read you work with interest, and found some important issues that require a moderate review. These issues are:

a) The title needs to be changed. This paper is really about ecological variability, not about coastline susceptibility as we know from several papers on the Eastern Mediterranean and Baltic Sea.

b) Your work is not entirely new. Please, refer to the outputs/papers resulting from the Nereids project (www.nereids.eu) sponsored by the EU, which have been published on Environmental Pollution, Scientific Reports, and Marine Pollution Bulletin in 2014, 2015 and 2016. These are important pieces of information and the first time extensive susceptibility mapping was achieved for the Eastern Mediterranean Sea, or for any other large sea in the world.

c) A main issue I have is why would someone use vulnerability coefficients in coastal areas. The rule in the case of oil spills as that oil is, by definition, harmful. I do not feel you need to take into account species' resilience to the oil spill until all pollution is clean, and you access the spill's impact. Therefore, if you acknowledge and summarise the work of Nereids above, you will be able to claim that you method is novel because it can be used to estimate ecological vulnerability to marine pollution post-spill.

d) The technical part of the paper is robust, but needs a further review of its English. Please, check also if the text needs changes pending the changes I suggest above. Also 'coastal zones' or 'coastlines' are better terms than 'sea-coastal zones'.

Sincerely yours

Tiago M. Alves

3D Seismic Lab - Cardiff University

Author Response

Round  2

Reviewer 2 Report

Dear authors

You seem to have ignored my previous review in all its totality. When I asked you to use Nereids' results in the paper was to establish a term of comparison with your work. In your reply letter, you just justified what should be in the text, i.e. that your analysis goes a step forward from Nereids' analyses to include an ecological variable to your study and maps.

It is also not polite to say that most maps are wrong, as stated in the Introduction. This fact, and the fact that you did not understand that previous publications (i.e.the state-of-the-art at present) should be mentioned in the introduction, justifies another round of reviews to this paper.

All in all, you misinterpreted my comments as a personal criticism and ended up including in your response letters parts (and concepts) that should be in the paper - in the Introduction and in the discussion - as a term of comparison with your work.
